# High-temperature concomitant metal-insulator and spin-reorientation transitions in a compressed nodal-line ferrimagnet Mn$_3$Si$_2$Te$_6$

Resta A. Susilo[1,11], Chang Il Kwon [1,2,11], Yoonhan Lee[3,11], Nilesh P. Salke [4], Chandan De [2], Junho Seo [1,2], Beomtak Kang[1,2], Russell J. Hemley [4,5,6], Philip Dalladay-Simpson [7], Zifan Wang[7], Duck Young Kim [7], Kyoo Kim[8], Sang-Wook Cheong [9,10], Han Woong Yeom [1,2], Kee Hoon Kim [3] ✉ & Jun Sung Kim [1,2] ✉

Symmetry-protected band degeneracy, coupled with a magnetic order, is the key to realizing novel magnetoelectric phenomena in topological magnets. While the spin-polarized nodal states have been identified to introduce extremely-sensitive electronic responses to the magnetic states, their possible role in determining magnetic ground states has remained elusive. Here, taking external pressure as a control knob, we show that a metal-insulator transition, a spin-reorientation transition, and a structural modification occur concomitantly when the nodal-line state crosses the Fermi level in a ferrimagnetic semiconductor Mn$_3$Si$_2$Te$_6$. These unique pressure-driven magnetic and electronic transitions, associated with the dome-shaped $T_c$ variation up to nearly room temperature, originate from the interplay between the spin-orbit coupling of the nodal-line state and magnetic frustration of localized spins. Our findings highlight that the nodal-line states, isolated from other trivial states, can facilitate strongly tunable magnetic properties in topological magnets.

Topological magnets where the symmetry-protected band degeneracy is coupled with magnetism have emerged as a new material platform for novel transport phenomena and spintronic functionalities[1–7]. Among various types of topological magnets, the so-called nodal-line magnetic semimetals or semiconductors are one of the most seminal examples that exhibit unprecedentedly large magnetotransport responses, including giant anomalous Hall effect (AHE)[8–14] and colossal

angular magnetoresistance (AMR)[15–17]. Unlike Dirac or Weyl magnets, the topological nodal-line magnets possess lines or loops of the band degeneracy in the momentum space, which can be lifted effectively by tunable spin-orbit coupling (SOC), producing strong Berry curvature with spin orientation. The key issue is then establishing a comprehensive picture of the intimate coupling between the magnetic and electronic degrees of freedom in the nodal-line magnets. While

[1]Department of Physics, Pohang University of Science and Technology, Pohang, Korea. [2]Center for Artificial Low Dimensional Electronic Systems, Institute for Basic Science (IBS), Pohang, Korea. [3]Department of Physics and Astronomy, CeNSCMR, Seoul National University, Seoul, Korea. [4]Departments of Physics, University of Illinois Chicago, Chicago, IL, USA. [5]Departments of Chemistry, University of Illinois Chicago, Chicago, IL, USA. [6]Department of Earth and Environmental Sciences, University of Illinois Chicago, Chicago, IL, USA. [7]Center for High Pressure Science and Technology Advanced Research, Shanghai, China. [8]Korea Atomic Energy Research Institute (KAERI), Daejeon, Korea. [9]Laboratory of Pohang Emergent Materials, Pohang Accelerator Laboratory, Pohang, Korea. [10]Rutgers Center for emergent Materials and Department of Physics and Astronomy, Rutgers University, New Brunswick, NJ, USA. [11]These authors contributed equally: Resta A. Susilo, Chang Il Kwon, Yoonhan Lee. ✉e-mail: optopia@snu.ac.kr; js.kim@postech.ac.kr

magnetic control of the electronic response of the nodal-line states has been demonstrated[8–15], the question of how the presence of the nodal-line states and their tuning affect the magnetic properties, such as the magnetic configurations and anisotropy, has rarely been addressed experimentally. The challenges are to find a system with the symmetry-protected nodal-line states in the vicinity of the Fermi level without trivial bands and to control them effectively with external perturbations.

The recently discovered nodal-line magnetic semiconductors can serve as a model system where the spin-polarized valence or conduction bands possess nodal-line band degeneracy. For such magnetic semiconductors, pressure offers a clean and continuous tuning parameter for modulating their electronic structures, e.g. band gap or width, and the magnetic exchange interactions without introducing disorders or doping[18]. Thus a pressure-driven magnetic transition or a metal-insulator transition (MIT) in nodal-line magnetic semiconductors can unveil the essential role of the nodal-line states in determining both magnetic and electronic properties. In this work, we address this issue by investigating magnetic, electronic, and structural properties of a nodal-line ferrimagnetic semiconductor $Mn_3Si_2Te_6$. Using the magnetotransport, magnetization and X-ray diffraction measurements at high pressures, we found that a pressure-driven MIT, a spin-reorientation transition (SRT), and a structural modification, occur concomitantly at a critical pressure $P_c \sim 14$ GPa. We also found an unusual dome-shaped ferrimagnetic transition ($T_c$) variation with pressure reaching up to nearly room temperature. These observations, together with the systematic variation of the AHE and AMR, reveal the critical role of the nodal-line states on the pressure-induced magnetic and electronic responses of $Mn_3Si_2Te_6$.

## Results and discussion

$Mn_3Si_2Te_6$ has a self-intercalated van der Waals structure with the tri-gonal $P\bar{3}1c$ space group, consisting of an alternating stack of the $MnSiTe_3$ layers with the hexagonal honeycomb network of $Mn_1$ atoms and the Mn layers with the triangular lattice of $Mn_2$ atoms (Fig. 1a)[19,20].

Electronically, it is a p-type narrow gap semiconductor and hosts an easy-plane (ab-plane) ferrimagnetic phase below $T_c \approx 78$ K with an antiparallel alignment of localized spins at the $Mn_1$ and $Mn_2$ sublattices (Fig. 1a). First-principles calculations revealed that this ferrimagnetic phase can be explained by magnetic frustration of competing anti-ferromagnetic (AFM) exchange couplings between the neighboring Mn spins[15,21]. When the uncompensated magnetization is rotated toward a hard axis (c-axis)[15,16], a huge AMR up to ~$10^{11}$%/rad, has been observed at low temperatures, named colossal AMR[9,15,16,21,22]. These remarkable properties have been attributed to a magnetic field-driven MIT due to the lifting of the nodal-line band degeneracy by a tunable spin-orbit coupling (SOC) gap with spin rotation[15]. Because the relevant electronic state is only the Te-derived valence bands with nodal-line band degeneracy, they are expected to approach the unoccupied band as the electronic gap ($\Delta$) shrinks under pressure, eventually inducing pressure-driven MIT (Fig. 1b). For the out-of-plane spin orientation, then the nodal-line states which hybridized with the Mn states of the localized spins can work as a strong source of Berry curvature nearby $E_F$ due to a finite SOC gap ($\Delta_{SOC}$) shown in Fig. 1c. Therefore $Mn_3Si_2Te_6$ serves as a model system to study the interplay of the nodal-line electronic states and the frustrated magnetic coupling.

The pressure effects on the electronic and magnetic properties of $Mn_3Si_2Te_6$ are clearly observed in its temperature-dependent ab-plane resistivity $\rho_{ab}(T)$ measured at different pressures (Fig. 1d, e). Upon increasing pressure, the overall magnitude of $\rho_{ab}(T)$ is continuously reduced, and the slope $d\rho_{ab}/dT$ changes from negative to positive across $P_c \sim 14$ GPa, signaling a pressure-driven MIT. Similar behavior was observed for six different crystals from three different batches (Supplementary Note 11). The crossover between the insulating and metallic behaviors is clearly separated by the Mott-Ioffe-Regel limit with $\rho_{MIR} = \hbar c/e^2$ (the dashed line in Fig. 1d), where c is the c-axis lattice constant of $Mn_3Si_2Te_6$. Accordingly, the activation behavior at low temperatures is dramatically suppressed (Supplementary Fig. S1), and the corresponding activation gap $\Delta$ extracted from the fits to the Arrhenius model closes at $P_c$ (Fig. 1f). The slow upturn of $\rho_{ab}(T)$ at low

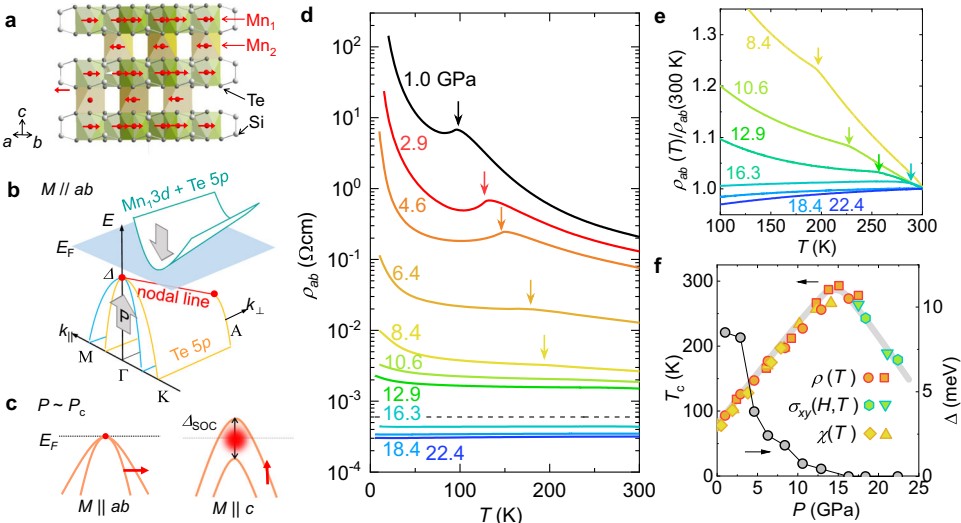

**Fig. 1 | Metal-insulator transition and high-$T_c$ ferrimagnetism of $Mn_3Si_2Te_6$.**
**a** Crystal structure of $Mn_3Si_2Te_6$. The $Mn_1$ and $Mn_2$ atoms are in the hexagonal and triangular lattices, respectively. The ferrimagnetic configuration is indicated by the (red) arrows. **b**, **c** Schematic illustrations of the semiconducting electronic structure. The Te 5p valence band with the nodal-line degeneracy and the conduction band with the hybridized $Mn$ 3d and Te 5p states become closer in energy with pressure (**b**). At the critical pressure $P_c$ for electronic gap closing, the lifting of topological band degeneracy and the associated Berry curvature (red dot) is determined by the spin-orbit coupling ($\Delta_{SOC}$) depending

on magnetization orientations, $M\|ab$ and $M\|c$ (**c**). **d** Temperature-dependent ab-plane resistivity $\rho_{ab}(T)$ of $Mn_3Si_2Te_6$ measured at different pressures. The resistive kink at $T_c$ shifts to higher temperatures with pressure indicated by the arrows. **e** The normalized $\rho_{ab}(T)$ with its room temperature value at high pressures. The kinks at $T_c$ is indicated by the arrows. **f** The dome-shaped $T_c$ variation up to nearly room temperature as a function of pressure, estimated from the ab-plane resistivity $\rho_{ab}$, the Hall conductivity $\sigma_{xy}$, and the magnetic susceptibility $\chi$. The corresponding activation gap $\Delta$ reduces with pressure and eventually closes at $P_c$.

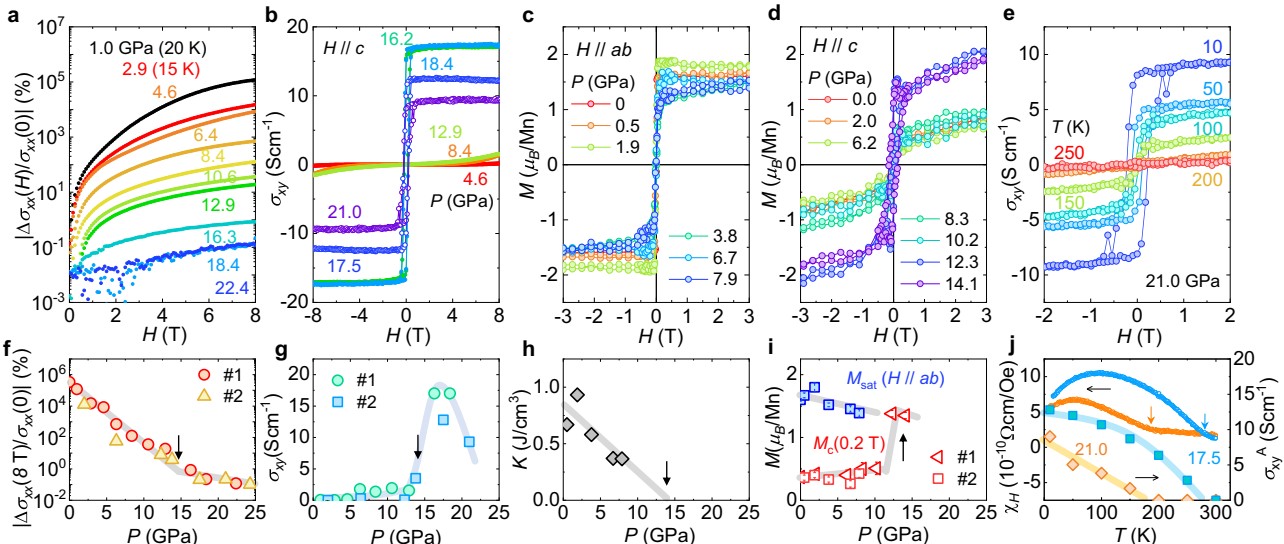

**Fig. 2 | Magnetotransport and magnetic properties of Mn$_3$Si$_2$Te$_6$ at high pressures. a** Magnetoconductivity $|\Delta\sigma_{xx}(H)/\sigma_{xx}(0)|$ of Mn$_3$Si$_2$Te$_6$ at high pressures up to ~22 GPa. The $|\Delta\sigma_{xx}(H)/\sigma_{xx}(0)|$ curves were taken at 5 K, otherwise specified in the parenthesis. **b** Magnetic field-dependent Hall conductivity $\sigma_{xy}(H)$ at different pressures, taken at 10 K. **c, d** Magnetic field-dependent magnetization $M(H)$ for $H\|ab$ (**c**) and $H\|c$ (**d**) at different pressures, taken at 5 K. **e** The representative $\sigma_{xy}(H)$ curves above $P_c$ ($P = 21.0$ GPa) at different temperatures. **f–i** Pressure-dependent magnetoconductivity $|\Delta\sigma_{xx}(8T)/\sigma_{xx}(0)|$ (**f**), Hall conductivity (**g**), the magnetocrystalline anisotropy energy $K$ (**h**), and the saturation magnetization $M_{sat}$ together with the low-field $c$-axis magnetization $M_c$ for $H\|c$ at 5 K. The critical pressure $P_c$ for the MIT is indicated by the vertical arrows in (**f–i**). **j** Temperature-dependent magnetic susceptibility $\chi_H(T)$, extracted from $\sigma_{xy}(H)/H$ data above $P_c$ ($P = 17.5$ and 21.0 GPa). The temperature showing a clear kink at $\chi_H(T)$, indicated by the arrows, is consistent with the onset of the anomalous Hall conductivity $\sigma_{xy}^A$ as a function of temperature.

temperatures above $P_c$ follows the logarithmic $T$ dependence, indicating the Kondo scattering[23] or disorder-induced localization[24,25]. The MIT at $P_c$ ~14 GPa is further confirmed by infrared (IR) reflectance spectroscopy, which reveals a drastic enhancement of the optical conductivity above ~$P_c$ (Supplementary Note 10). These results are consistent with the recent results in ref. 26, but not in ref. 27, most likely due to different doping levels in Mn$_3$Si$_2$Te$_6$ crystals (Supplementary Note 11).

In addition to the MIT, we observed a significant change of the resistivity anomaly, corresponding to a ferrimagnetic transition at $T_c$[15,16,21]. This resistive anomaly shifts towards higher temperatures with pressure, reaching nearly room temperature at $P_c$ ~14 GPa, as seen more clearly in the normalized resistive curves $\rho(T)/\rho(300\text{ K})$ (Fig. 1e). In the metallic regime above $P_c$, the resistivity anomaly is no longer observable. However, as discussed below, $T_c$ can be traced by the AHE measurements, which clearly decreases with pressure above $P_c$ (Fig. 2). Consistently, temperature-dependent magnetic susceptibility $\chi(T)$ on Mn$_3$Si$_2$Te$_6$ confirms strong enhancement of $T_c$ under pressure for both $H\|ab$ and $H\|c$, as the onset of $\chi(T)$ shifts to higher temperatures (Supplementary Fig. S4). This behavior is also consistent with recent reports on the enhancement of $T_c$ with pressure[27,28]. The pressure-dependent $T_c$ data estimated from $\rho_{ab}(T)$ and $\chi(T)$ match well with each other, firmly constructing a dome-shaped $T_c$ variation with a maximum $T_c$ close to room temperature (Fig. 1f).

The magnetotransport properties of Mn$_3$Si$_2$Te$_6$ also exhibit systematic changes with pressure. The temperature-dependent $\rho_{ab}(T)$ curves at different magnetic fields and pressures reveal that the magnetoconductivity (MC) becomes weaker in the more metallic state at higher pressures (Supplementary Fig. S2). This is clearly seen in the field-dependent MC, defined as $\Delta\sigma(H)/\sigma(0)$ for $H\|c$ at different pressures (Fig. 2a). As demonstrated at ambient pressure[15], $\Delta\sigma(H)/\sigma(0)$ is dominated by the AMR with rotating magnetization toward the $c$-axis under $H\|c$. Because the key mechanism is the lifting of the nodal-line degeneracy of the spin-polarized valence bands due to spin-orbit coupling ($\Delta_{SOC}$) and the resulting closure of the electronic gap ($\Delta$), the relative sizes of $\Delta$ and $\Delta_{SOC}$ are the main parameters determining the

AMR. As illustrated in Fig. 1b, the reduction of the electronic gap $\Delta$ results in the semimetallic band structures at high pressures. Assuming that the SOC gap $\Delta_{SOC}$ remains nearly the same, the AMR is expected to be suppressed with pressure, which is indeed what is observed experimentally (Fig. 2f). Upon increasing pressure, $\Delta\sigma(H)/\sigma(0)$ at 20 K drops by four orders of magnitude and becomes negligible above $P_c$ ~14 GPa. Therefore significant reduction of the AMR, together with suppression of the activation gap estimated from $\rho_{ab}(T)$ (Fig. 1f), strongly suggests that the nodal-line bands approach and eventually cross the Fermi level $E_F$ at a critical pressure $P_c$.

This conclusion is further supported by the Hall response of Mn$_3$Si$_2$Te$_6$ with pressure. In a low-pressure region ($P < 13$ GPa), the Hall conductivity $\sigma_{xy}(H)$ shows a non-linear field-dependence with an initial exponential increase at low magnetic fields (Supplementary Fig. S3a). Such an unusual behavior of $\sigma_{xy}(H)$ is distinct from the field-dependent magnetization $M(H)$ and the conventional AHE, but it can be understood by considering the strong field-dependence of the activation gap and thus the density of hole carriers (Supplementary Note 2). When Mn$_3$Si$_2$Te$_6$ enters the metallic phase above $P_c$, $\sigma_{xy}(H)$ shows a qualitatively different behavior with a large jump of $\sigma_{xy}(H)$ and a clear magnetic hysteresis at low magnetic fields. These features are the hallmarks of the AHE of ferro- or ferrimagnets with perpendicular magnetic anisotropy[29], suggesting the pressure-driven spin reorientation in Mn$_3$Si$_2$Te$_6$ across $P_c$. With the spontaneous spin alignment along the $c$-axis, the lifted nodal-line degeneracy produces strong Berry curvature (Fig. 1c), resulting in a large AHE. Consistently, the $\sigma_{xy}(H)$ value at $H = 1$ T increases substantially at $P_c$ and then weakens slightly with further increasing pressure (Fig. 2g). These results thus support the fact that the nodal-line bands cross the $E_F$ when the MIT occurs. Using this significant AHE above $P_c$, we tracked the $T_c$ of Mn$_3$Si$_2$Te$_6$. The magnetic susceptibility, estimated from $\chi_H - \sigma_{xy}(H)/H$, and the anomalous Hall conductivity $\sigma_{xy}^A(0)$ by remnant magnetization shows that $T_c$ remains close to the room temperature just above $P_c$, but shifts to lower temperatures down to 200 K at 21.0 GPa (Fig. 2e, j). The opposite pressure dependence of $T_c$ below and above $P_c$ leads to a dome-shaped phase boundary, separated by the pressure-driven MIT.

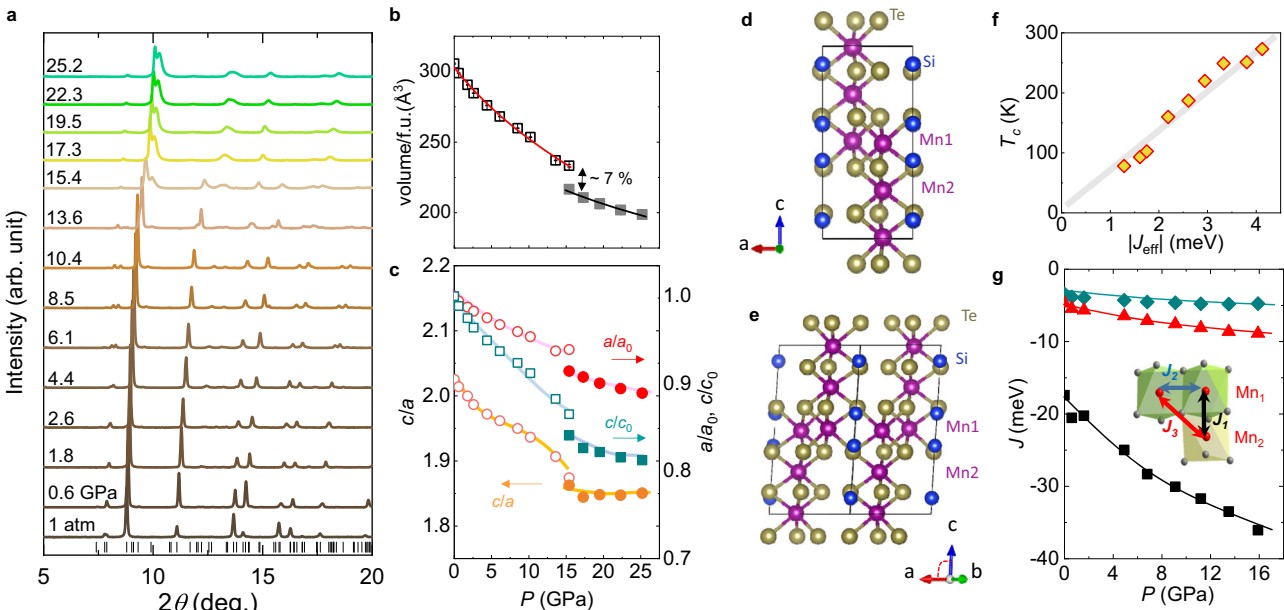

**Fig. 3 | Structural modification of $Mn_3Si_2Te_6$ at high pressures. a** Synchrotron X-ray diffraction patterns of $Mn_3Si_2Te_6$ at different pressures up to 25.2 GPa, with the vertical bars indicating the Bragg markers for the ambient pressure phase. **b, c** Pressure dependence of the unit cell volume (**b**) and lattice parameters (**c**). The fits using the third-order Birch−Murnaghan equation of state are presented with solid lines in (**b**) both for the low and high-pressure phases. The normalized $a$ and $c$ parameters for the high-pressure phase are defined in the pseudo-hexagonal lattice for direct comparison with the low-pressure phase. The $c/a$ ratio is also shown in

(**c**), revealing distinct pressure-dependent responses below and above $P_c$. **d, e** The structures of low (**d**) and high pressure (**e**) phases of $Mn_3Si_2Te_6$. **f** Experimental $T_c$ as a function of the effective exchange interaction strength $|J_{eff}| = |J_3 - J_2|$, showing a clear liner relationship between them below $P_c$. **g** Calculated nearest-neighbor exchange couplings at different pressures. Three nearest-neighbor exchange interactions $J_i$ ($i = 1, 2, 3$) between the localized Mn spins of the face-sharing, edge-sharing, and corner-sharing octahedra, responsible for the magnetic ground state of $Mn_3Si_2Te_6$ below $P_c$.

The isothermal magnetization measurements at high pressures confirm the pressure-driven SRT. Figure 2c, d show magnetic field-dependent $M(H)$ at 5 K measured under different pressures for $H\|ab$ and $H\|c$, respectively. At ambient pressure, the $ab$-plane magnetization $M_{ab}$ spontaneously increases and reaches a saturation magnetic moment $M_{sat}$ of ~$1.7\mu_B$ at a low field of $H$ ~0.2 T, while the $c$-axis magnetization $M_c$ sharply increases at low fields but then slowly rises up to $M_{sat}$ at higher fields. The larger saturation field $H_{sat}$ for $H\|c$ than for $H\|ab$ indicates the easy-plane ($ab$) magnetic anisotropy, and the corresponding magnetic anisotropy energy scale $K$ can be estimated using the equation $H_{sat}^c = 2K/M_{sat}$. By extrapolation of $M(H)$ data to higher fields for both $H\|c$ and $H\|ab$ (Supplementary Note 3 and Fig. S4), we estimated the pressure-dependent $K$, which clearly decreases with pressure and is expected to change its sign around $P_c$ ~14 GPa (Fig. 2h). At higher pressures for $H\|c$ (Fig. 2d) a sudden increase of the low-field magnetization is observed near $P_c$, accompanied by a clear hysteresis loop, which indicates the perpendicular magnetic anisotropy. The $M(H)$ data can be nicely reproduced by $\sigma_{xy}(H)$ above $P_c$ (Fig. 2b), following a relation $\sigma_{xy}(H) = S_H M(H)$ (Supplementary Note 2 and Fig. S3b). After the SRT transition, the low-field magnetic moment for $H\|c$ at $H = 0.2$ T reaches ~$1.5\mu_B/$Mn, following the trend of $M_{sat}$ at lower pressures (Fig. 2i). With further increasing magnetic field, $M_c$ slowly increases up to ~$2\mu_B/$Mn, which remains close to the value for the high-spin ferrimagnetic phase ($t_{2g}^3 e_g^2$, ~5/3 $\mu_B/$Mn), clearly distinguished from the values expected for the ferromagnetic phases of the high-spin (~$5\mu_B/$Mn) and the low-spin ($t_{2g}^5 e_g^0$, ~1 $\mu_B/$Mn) states of $Mn^{2+}$ atoms. These observations suggest that neither the spin crossover transition from the high-spin to the low-spin states nor the ferrimagnetic-to-ferromagnetic transitions occur in $Mn_3Si_2Te_6$ at high pressure. Rather, the SRT from the easy-plane to easy-axis magnetic anisotropy concomitantly occurs within the ferrimagnetic phase at $P_c$.

In addition to the MIT and SRT, a structural modification occurs at $P_c$. The synchrotron X-ray diffraction patterns (XRD) of $Mn_3Si_2Te_6$ up

to ~25 GPa (Fig. 3a) show that up to $P_c$ ~14 GPa, all the diffraction peaks are identified with the trigonal structure with $P\bar{3}1c$ space group, same as the ambient structure. However, the above $P_c$ X-ray diffraction patterns are suddenly modified, indicating a structural transformation. Although the accurate determination of the crystal structure is known to be problematic due to sample strain and texture at high pressures[18,30], we identified several structural models that can produce good fitting to the XRD data (Supplementary Note 5) and then compared their total energies from the DFT calculations. Among them, we found that a monoclinic structure with the $C2/c$ space group has the lowest total energy and, thus, is likely the candidate structure above $P_c$ (Supplementary Table S1). The monoclinic high-$P$ phase can be described as a distorted trigonal $P\bar{3}1c$ structure in which a single Te site is split into three distinct sites and a weak sliding of the $MnSiTe_3$ layers relative to each other induces a slight monoclinic distortion by ~3°. Despite the symmetry lowering, the structural motif remains the same in both low- and high-$P$ phases i.e., the alternating stack of the $MnSiTe_3$ layers and the triangular lattice remains in the high-pressure phase (Fig. 3d, e). The unit cell volume $V$ derived from the Le Bail fitting to the XRD patterns vary smoothly with pressure below $P_c$, well fitted with a third-order Birch−Murnaghan equation of state[31,32] (Fig. 3b), and then shows a mild drop of ~7% at $P_c$. By describing the lattice parameters above $P_c$ into the pseudo-trigonal lattice system, $a = b = \frac{1}{2}\sqrt{a_m^2 + b_m^2}$, $c = c_m$ ($a_m$ and $c_m$, the lattice parameters for the $C2/c$ structure - see Supplementary Note 5), we found that this mild volume reduction at $P_c$ is due to small shrinkage in both the in-plane and the out-of-plane lattice parameters by ~2% across $P_c$ (Fig. 3b, c). The pressure-dependent lattice parameters ($a$, $c$) show that the system is more compressible along the $c$-axis than the $a$-axis below $P_c$, consistent with the pressure-dependent Raman spectra (Supplementary Note 4), whereas the $ab$-plane and the $c$-axis are compressed nearly isotropic with almost constant $c/a$ ratio above $P_c$. This distinct lattice response is

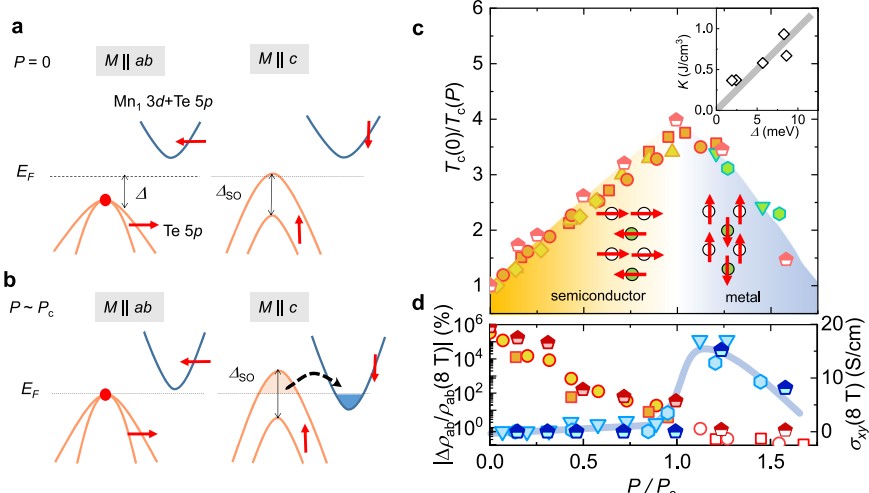

**Fig. 4 | Electronic structures and phase diagram of Mn₃Si₂Te₆ under pressures.**
**a, b** Schematic illustration of the electronic structure at ambient (**a**) and critical (**b**) pressures for the in-plane ($M\|ab$) and the out-of-plane ($M\|c$) spin orientation. The valence Te $5p$ bands with nodal-line degeneracy (red dot) are lifted by spin rotation due to the SOC gap $\Delta_{SO}$. At ambient pressure, spin rotation by an external magnetic field along the $c$-axis leads to the MIT (**a**). At a critical pressure $P_c$ for the pressure-driven MIT (**b**), spin rotation to the $c$-axis induces charge transfer from the valence Te $5p$ bands to the conduction Mn₁ $3d$/Te $5p$ hybridized bands, which provides an additional channel for lowering the total energy for $M\|c$, leading to the spin-reorientation transition. **c** Pressure-dependent magnetic phase diagram of both undoped and doped Mn₃Si₂Te₆ as a function of reduced pressure ($P/P_c$), associated with the pressure-driven MIT, SRT, and a dome-shaped $T_c$ variation. The critical pressures, $P_C$, are estimated to be 14.5 and 15.5 GPa for the undoped and doped samples, respectively. The inset shows a linear relationship between the magnetocrystalline anisotropy energy $K$ and the activation energy $\Delta$ below $P_c$. **d** Pressure-dependent magnetoresistance and Hall conductivity. Half-filled symbols in (**c**, **d**) represent the data for the doped sample.

possibly related to the opposite trend of $T_c$ across $P_c$, as discussed below.

The ferrimagnetic ground state of Mn₃Si₂Te₆ is the consequence of magnetic frustration between three types of the nearest-neighbor superexchange coupling $J_i$ ($i = 1, 2, 3$) (Fig. 3g). All AFM exchange couplings between Mn spins of the face-sharing ($J_1$), of the edge-sharing ($J_2$), and of the corner-sharing ($J_3$) octahedra compete with each other with a hierarchy of $J_1 > J_3 > J_2$, as confirmed by first principle calculations and neutron scattering[15,21,33]. This hierarchy of $J$'s makes the intralayer FM configurations of Mn₁ spins and the interlayer AFM configurations of Mn₁ and Mn₂ spins, which stabilizes the ferrimagnetic phase. Upon increasing pressure, these exchange couplings remain AFM and become larger in magnitude, as estimated from the first principle calculations (Fig. 3g). The largest interlayer $J_1$ coupling shows a much faster increase with pressure than the others because the $J_1$ direct $d$-$d$ exchange interaction is significantly enhanced by shorting the interlayer Mn₁-Mn₂ distance due to the strong lattice contraction along the $c$-axis (Fig. 3c). On the other hand, the anisotropic lattice response increases the Mn₁-Te-Mn₂ angle of the corner-sharing octahedra and thus the $J_3$ coupling, whereas the $J_2$ coupling within the layers is enhanced mildly due to the relatively weak reduction of the Mn₁-Mn₁ distance (Supplementary Fig. S9). Since $J_3$ grows faster than $J_2$ with pressure below $P_c$, the difference between them, $J_{eff} \sim |J_3 - J_2|$, increases significantly with pressure, inducing a linear increase of the measured $T_c$ with $J_{eff}$ (Fig. 3f). In the high-pressure phase above $P_c$, however, the compressibilities of the $ab$-plane and $c$-axis become similar to each other (Fig. 3c), which is more effective in enhancing $J_2$ by making the AFM direct exchange stronger with a shortened intralayer Mn₁-Mn₁ distance. As a result, the difference between $J_2$ and $J_3$, i.e., $J_{eff}$, is expected to be reduced, which promotes magnetic frustration and thereby suppresses $T_c$ above $P_c$ (Fig. 1f). These observations suggest that different lattice responses to pressure, which tips over the subtle balance of frustrated magnetic couplings, have a solid connection to the strong and opposite pressure dependence of $T_c$ across $P_c$.

Having established that the MIT, the SRT, and structural modification occur at ~$P_c$, we now discuss the possible origin of such concomitant transitions in Mn₃Si₂Te₆. In various manganese chalcogenides, including MnP$Ch_3$[34], Mn$Ch$[35] and Mn$Ch_2$ ($Ch$ = S, Se, Te)[36–39], the pressure-driven MIT is known to be accompanied by a structural transition. In these manganese chalcogenides, the spin crossover transition from the high-spin ($t_{2g}^3 e_g^2$) to the low-spin ($t_{2g}^5 e_g^0$) states of Mn²⁺ atoms commonly triggers the so-called giant volume collapse by at least ~20% and introduces the Mn-Mn metallic bonding due to the drastic reduction of the Mn²⁺ ionic size from 0.83 to 0.67 Å. In contrast, the ferrimagnetic phase in a high-spin Mn²⁺ state of Mn₃Si₂Te₆ remains stable across $P_c$, accompanied by a moderate volume reduction of ~7%. This clearly distinguishes its pressure-driven transition from those found in the other manganese chalcogenides (Supplementary Table S2). Furthermore, the pressure-driven MIT, associated with the SRT, is also observed in 20% Se-doped Mn₃Si₂Te₆ crystal, showing qualitatively the same behaviors of the magneto-transport properties and the dome-shaped $T_c$ variation with pressure (Supplementary Note 7). Considering the different structural parameters which lead to different bond lengths and angles in the undoped and doped crystals, these observations imply that, unlike the other manganese chalcogenides, the pressure-driven modulation of the electronic structure plays a critical role in triggering the other concomitant transitions in Mn₃Si₂Te₆.

The key difference of the pressure-driven MIT in Mn₃Si₂Te₆ from other manganese chalcogenides is the touching of the nodal-line states at $E_F$, which is confirmed by the drop of the AMR and the sudden rise of the AHE at $P_c$ (Fig. 2a, b). In this case, the strong SOC of the nodal-line states can induce the SRT. At ambient pressure, the localized Mn spins of Mn₃Si₂Te₆ prefer the in-plane alignment due to the magnetic exchange anisotropy[40]. The spin-polarized nodal-line states of the Te-rich valence bands, well below the $E_F$, are mostly occupied regardless of the spin orientation, which makes a minor contribution to the magnetic anisotropy (Fig. 4a). However, when the degenerate nodal-line states are located close to the $E_F$ near $P_c$, the lifting of the band degeneracy by strong SOC with the out-of-plane spin orientation,

parallel to the orbital angular momentum of the Te bands, can push the two split bands well above and below the $E_F$ (Fig. 4b). As a consequence, charge transfer from the higher-energy SOC-split band to the lower-energy conduction band provides an energy gain for the perpendicular magnetic anisotropy. The strong correlation between the magnetic anisotropic energy $K$ and the activation gap $\Delta$ is consistent with this picture (Fig. 4c). Furthermore, transport and Raman spectroscopy measurements taken on decompression indicate that the MIT and SRT are reversible with significant hysteresis (Supplementary Note 9). Even in the presence of substantial strain disorder in the decompressed crystal, the preferred spin orientation, either in the in-plane or along the out-of-plane, is well defined and correlated with the electronic states, either gapped or gapless. Therefore, it can be concluded that, indeed, the MIT and SRT are tied together due to the SOC effect in the nodal-line states.

The pressure-driven instability for the SRT and the resulting interband charge transfer also provide a reasonable explanation for the structural modification at $P_c$. In $Mn_3Si_2Te_6$, the lowest-energy conduction bands are mainly from the in-plane $Mn_1$ $3d$ orbitals hybridized Te $5p$ orbitals. Below $P_c$, the in-plane contraction is mainly related to the displacement of the Te atoms in the $MnTe_6$ octahedra (Supplementary Fig. S9), leading to a relatively small contraction in the $ab$-plane than along the $c$-axis. At $P_c$, however, the occupation of the bonding state of $Mn_1$ orbitals due to the interband charge transfer is effective in reducing the intralayer $Mn_1$-$Mn_1$ distance, and therefore the in-plane lattice parameter, in good agreement with the sudden shrinkage of the $ab$-plane (Fig. 3c). Above $P_c$, the semimetallic band structure promotes intralayer and interlayer metallic bonding, making a more isotropic lattice response to pressure with a nearly constant $c/a$ ratio with pressure (Fig. 3c). The MIT with the nodal-line states and their strong SOC provide a natural explanation for the concomitant SRT and structural transition at $P_c$.

Our findings establish a concrete example in which the spin-polarized nodal-line states are actively involved in determining the magnetic properties in topological magnets. This exceptional tunability of the magnetic and electronic properties with pressure is a consequence of several attributes of $Mn_3Si_2Te_6$. The magnetic frustration with competing AFM coupling channels makes $T_c$ extremely sensitive to pressure-induced electronic and structural modification. The high-spin configuration ($t_{2g}^3 e_g^2$) of $Mn^{2+}$ atoms suppresses the single-ion anisotropy without orbital degrees of freedom, which makes the system more susceptible to the Te states with strong SOC. The absence of other trivial electronic states at $E_F$ is also crucial for the nodal-line state with strong SOC to contribute significantly to magnetic and electronic properties. These key ingredients and their interplay enable to trigger of unprecedented pressure-driven phase transitions in $Mn_3Si_2Te_6$, providing the material-guiding principle for small-gap magnetic semiconductors with topological band degeneracy. Our findings demonstrate that tuning the energy position of the isolated nodal-line states with respect to the Fermi level by e.g., strain or electrical gating offers a novel route for the spin-related functionalities in magnetic semiconductors.

## Methods

### Single crystal growth
Single crystals of $Mn_3Si_2Te_6$ were grown using a high-temperature self-flux method[15]. A mixture of Mn (99.95 %), Si (99.999 %), and Te (99.999 %) in a molar ratio of 1:2:6 was placed in an alumina crucible, and another empty alumina crucible was kept on top of it with quartz wool separation. The whole crucible assembly was sealed in an evacuated quartz ampoule and first heated in a muffle furnace up to 1000 °C in 12 h and kept dwelling for 24 h to obtain a homogeneous solution. The furnace was then slowly cooled down to 700 °C in 150 h and held for 12 h at 700 °C. The ampoule was then quickly taken out and centrifuged to separate the crystals from the fluxes.

A standard chemical vapor transport (CVT) technique was employed to grow the 20% Se-doped crystals[15]. $I_2$ was used as a transport agent. A temperature gradient of 750 to 700 °C was maintained for 400 h for the crystal growth followed by cooling to room temperature at 70 °C h$^{-1}$.

### High-pressure magnetotransport and magnetization experiments
The resistivity data at high pressure were measured up to ~23 GPa in Quantum Design PPMS using a non-magnetic diamond anvil cell (DAC) made of a NiCrAl alloy. The size of the diamond culet used was 400 μm in diameter. $Mn_3Si_2Te_6$ single crystal was cut into a square of ~80 μm in width with 10 μm in thickness. NaCl was used as a pressure medium for all runs. We used the van der Pauw four-probe method to measure electrical resistance by using platinum (Pt) foil as electrodes. Two ruby balls were put inside the sample chamber to determine the pressure[41]. Magnetization measurements under pressure were performed in a Quantum Design MPMS using a non-magnetic miniature DAC made of Cu-Be alloy with the diamond anvils culet of 600 μm. A $Mn_3Si_2Te_6$ crystal of about 150 μm × 150 μm × 30 μm in size was loaded in a sample chamber made by a laser-drilled Rhenium gasket with silicon oil as pressure transmitting medium and ruby as pressure calibrant[41].

### Synchrotron X-ray diffraction experiments at high pressures
Synchrotron X-ray diffraction (XRD) measurements were conducted at beamline 16-BM-D, Sector 16, HPCAT at the Advanced Photon Source, Argonne National Laboratory ($\lambda$ = 0.4833 Å). Several small single crystals of $Mn_3Si_2Te_6$ were ground into powder and loaded into the diamond anvil cell. Argon was used as a pressure transmitting medium, and ruby was used to determine pressure[41]. A small grain of gold powder located inside the sample chamber was also used as an additional pressure determinant via the known EOS[42]. The 2D diffraction images were integrated using DIOPTAS software[43]. The Le Bail fitting to the X-ray diffraction patterns were accomplished using the Fullprof/Winplotr software[44,45]. Despite the use of different pressure medium between different sets of experiments, the MIT, SRT, and structural modification occur at a similar critical pressure $P_c$ ~14 GPa. This indicates that $Mn_3Si_2Te_6$ is not sensitive to the different hydrostatic conditions.

### First-principles calculations
Density functional theory calculation within project augmented wave method[46] was performed utilizing Vienna ab initio package (VASP)[47]. The Perdew-Burke-Ernzerhof exchange-correlation functional[48] is used, and the relativistic effect is considered within the second variational spin-orbit interaction scheme. To take into account of Coulomb correlation effect, DFT+U calculation within Dudarev scheme[49] with effective Coulomb potential of $U$ = 3 eV in Mn d orbital have been used. The plane energy cutoff of 450 eV and $k$ mesh of 12 × 6 × 12 for the Brillouin zone integration is used for the calculation. To compare the total energy of the experimentally suggested high-pressure structure candidates, we adopted the experimental lattice constant and optimized the internal positions of each atoms.

## Data availability
All data supporting the findings of this study are available within the main text and the Supplementary Information file. The data that support the findings of this study are available from the corresponding authors upon reasonable request. Source data are provided with this paper.

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

## Acknowledgements

This work was supported by the Basic Science Research Program (NRF-2022R1A2C3009731, RS-2023-00272090), BrainLink program (No. 2022H1D3A3A01077468), the Max Planck POSTECH/Korea Research Initiative (Grant No. 2022M3H4A1A04074153 and 2020M3H4A2084417),

funded by the Ministry of Science and ICT through the National Research Foundation (NRF) of Korea. R.A.S. was supported by the Basic Science Research Program through the National Research Foundation of Korea (NRF) funded by the Ministry of Education (Grant No. 2021R1I1A1A01060209). This work is also supported by the Institute for Basic Science (IBS) through the Center for Artificial Low-Dimensional Electronic Systems (no. IBS-R014-D1). The work at SNU was financially supported by the Ministry of Science and ICT through the National Research Foundation of Korea (No. 2019R1A2C2090648), by the Ministry of Education (No. 2021R1A6C101B418), and the National Research Foundation of Korea (NRF) grant funded by the Korea government (MSIT) (No. RS-2023-00220471). D.Y.K and Z.W. acknowledge the financial support from the National Natural Science Foundation of China (11774015, U1930401). K.K. was supported by the internal research and development program at KAERI (Grant 524550-24). S.-W.C. was supported partially by the DOE under Grant No. DOE: DE-FG02-07ER46382. N.P.S. and R.J.H. were supported by DOE-SC (DE-SC0020340), DOE-NNSA (DE-NA0003975), and NSF (DMR-2119308). Portions of this work were performed at HPCAT (Sector 16), Advanced Photon Source (APS), Argonne National Laboratory. HPCAT operations are supported by DOE-NNSA's Office of Experimental Sciences. The Advanced Photon Source is a U.S. Department of Energy (DOE) Office of Science User Facility operated for the DOE Office of Science by Argonne National Laboratory under Contract No. DE-AC02-06CH11357.

## Author contributions

J.S.K. and R.A.S. conceived the project. The single crystals were grown by C.D., J.S., B.K., H.W.Y. and S.-W.C. High-pressure transport experiments were carried out by R.A.S. and C.I.K. High-pressure magnetic measurements were performed by Y.L. and K.H.K. C.I.K. performed the Raman spectroscopy measurements. Synchrotron X-ray diffraction experiments were carried out by N.P.S. and R.J.H. First-principles calculations were performed by K.K. Z.W., and D.Y.K. P.D.-S. carried out the infrared spectroscopy measurements. R.A.S., C.I.K., Y.L., K.H.K. and J.S.K. performed data analyses and co-wrote the manuscript. All authors discussed the results and commented on the paper.

## Competing interests

The authors declare no competing interests.
