## [Peer Review File · Nature Communications]

Reviewers' Comments:

Reviewer #1:

Remarks to the Author:

The authors have investigated magnetic, electronic, and structural properties of a nodal-line ferrimagnetic semiconductor $\text{Mn}_3\text{Si}_2\text{Te}_6$. Using the magnetotransport, magnetization and X-ray diffraction measurements at high pressures, they found that a pressure-driven metal-insulator transition (MIT), a spin-reorientation transition (SRT), and a structural modification, occur concomitantly at a critical pressure $P_c \sim 14$ GPa. Based on these observations, the authors claimed that the nodal-line states play critical role on the pressure-induced magnetic and electronic responses of $\text{Mn}_3\text{Si}_2\text{Te}_6$. Although the manuscript is well presented, I do not agree with the interpretations raised by the authors.

1. The main observations are very similar to and different from that reported in Ref. 22, including the pressure-driven MIT and structural modification. In ref. 22, Wang et al. reported that the pressure-driven MIT occurs between 1.5 and 2.5 GPa, but the structural modification takes place at ~ 12 GPa. Moreover, in a recent theoretical work (PHYSICAL REVIEW B 106, 045106 (2022)), the DFT calculations indicate that the critical pressure of the MIT depends on the value of LSDA + SOC + U_{eff} . Based on the LSDA + SOC + $U_{\text{eff}} = 0.5$ eV calculations, Zhang et al. found that the critical pressure for the MIT to be about 2.4 GPa, close to the experimental observation (1.5–2.5 GPa).

2. The main highlight of this manuscript could be the claim that the pressure-induced transitions occur concomitantly when the nodal-line state crosses the Fermi level in the ferrimagnetic semiconductor $\text{Mn}_3\text{Si}_2\text{Te}_6$. However, from ref. 22 and the DFT calculations, it is clear that the MIT could be decoupled with both SRT and structural modification. Moreover, the SRT can also be explained by the structural modification.

Based on these considerations, I do not support this manuscript to be published in NC.

Reviewer #2:

Remarks to the Author:

In this manuscript, Susilo et. al. studied the transport, magnetic properties, and crystal structure of the ferrimagnetic semiconductor $\text{Mn}_3\text{Si}_2\text{Te}_6$ under high pressures by using the diamond anvil cell. This work reveals that the pressure-induced metal-insulator transition is accompanied with the change of magnetic and crystal structures, undergoing a pressure-induced spin reorientation in the metallic state above the critical pressure P_c . The authors highlight the tuning of pressure on the nodal-line states and the physical properties of $\text{Mn}_3\text{Si}_2\text{Te}_6$. In my opinion, this work is of high quality and deserves publication after addressing the following issues:

1) The authors can easily determine the T_c at lower pressures by using the kink feature in resistivity. Can the authors use the same criteria to define T_c above P_c in the metallic state?

2) The authors stated that the evolution of T_c in this work is consistent with previous reports, e.g. Ref. 26, Phys. Rev. B 106, 045106 (2022). However, the behaviors of resistance are very different from that of Ref. 26. What is the cause for such differences?

3) As described in the methods, the authors have used different pressure transmitting media for different high-pressure measurements. However, the results from different measurements seem to be quite consistent. Are there any influence of different medium on the physical properties? If so the authors may discuss briefly.

4) The authors stated that a pressure-driven spin reorientation appears across the P_c , which can be extracted from the $M(H)$ curves with obvious hysteresis loop for $H//c$ at higher pressures above 12GPa. While for $H//ab$, the $M(H)$ is only measured to 7.9GPa, as is shown in Fig.2. Why the authors only increase the pressure to 7.9GPa? They should provide the $M(H)$ curves at higher pressures to further confirm the pressure-induced spin reorientation with obvious reduction of magnetization for $H//ab$.

5) As indicated by the high-pressure XRD patterns, a structural transformation appears at the P_c from trigonal structure to the monoclinic structure. However, the authors discuss the changes of electronic structures with the same crystal structure below and above the P_c .

6) As reported in the previous report Phys. Rev. B 106, 045106 (2022), the high-pressure phase can be quenched to lower pressures. Have the authors measured the transport properties, magnetization and high-pressure XRD in the decompression process? If the transition is irreversible, the authors can measure the magnetic properties at even ambient pressure and can

further confirm the observed results.

Q1-1. The main observations are very similar to and different from that reported in Ref. 22, including the pressure-driven MIT and structural modification. In ref. 22, Wang *et al.* reported that the pressure-driven MIT occurs between 1.5 and 2.5 GPa, but the structural modification takes place at ~ 12 GPa. Moreover, in a recent theoretical work (PHYSICAL REVIEW B 106, 045106 (2022)), the DFT calculations indicate that the critical pressure of the MIT depends on the value of LSDA + SOC + U_{eff} . Based on the LSDA + SOC + $U_{\text{eff}} = 0.5$ eV calculations, Zhang *et al.* found that the critical pressure for the MIT to be about 2.4 GPa, close to the experimental observation (1.5–2.5 GPa).

A1-1. We appreciate the helpful comments from the reviewer. As the reviewer pointed out, there is a clear deviation between our results and those of Wang *et al.* [Phys. Rev. B, 106, 045106 (2022)]. As described in detail below, we have conducted additional transport experiments on several different crystals, as well as infrared (IR) spectroscopy experiments. All the data consistently show that the metal-insulator transition (MIT) occurs at ~ 13 -14 GPa, which is also in good agreement with the recent work conducted by a different group [arXiv: 2309.05945 (2023)]. They claimed that MIT occurs above 10 GPa and certainly not at $P \sim 1.5$ - 2.5 GPa as claimed by Wang *et al.* based on a single measurement on a single sample [Phys. Rev. B, 106, 045106 (2022)].

The critical pressure for the MIT at ~ 13 -14 GPa as determined in our work, relies on measurements for six different $\text{Mn}_3\text{Si}_2\text{Te}_6$ crystals from three different batches. We employed a standard criterion to determine the MIT critical pressure, where the system is inferred to be metallic when the slope of resistivity shows a positive temperature dependence ($d\rho/dT > 0$). The crossover between insulating and metallic behaviors is clearly separated by the Mott-Ioffe-Regel (MIR) limit *i.e.* $\rho_{\text{MIR}} = \hbar c/e^2$ (depicted by the dashed line in Fig. R1a) where c is the c -axis lattice constant (~ 14 Å) of $\text{Mn}_3\text{Si}_2\text{Te}_6$. As presented in Fig. R1a, the pressure variation of

FIG. R1. (a) Pressure dependence of room temperature in-plane resistivity of $\text{Mn}_3\text{Si}_2\text{Te}_6$ crystals studied in this work. For comparison, we included the resistivity value estimated from the optical conductivity at low frequency $\sigma(\omega = 600 \text{ cm}^{-1})$ which is consistent with the transport data. Dashed line represents the estimated Mott-Ioffe-Regel limit. (b) Comparison of the pressure-dependent in-plane resistivity, normalized by its value at $P \sim 1$ GPa, for different $\text{Mn}_3\text{Si}_2\text{Te}_6$ crystals.

room-temperature resistivity demonstrates excellent agreement between our different measurements with all data collapsing into a single curve. Additionally, we observed that the pressure-dependence resistivity estimated from the optical conductivity (see below) at low frequency, $\sigma(\omega=600 \text{ cm}^{-1})$, is consistent with the transport results. Furthermore, the recent study by Huang *et al* [arXiv: 2309.05945 (2023)] claimed that MIT occurs above 10 GPa, which agrees well with our work. As shown in Fig. R1b, the pressure-dependence of normalized resistance $\rho_{ab}(P)/\rho_{ab}(\sim 1 \text{ GPa})$ by Huang *et al* [arXiv: 2309.05945 (2023)] follows a similar trend to our results albeit exhibiting a slightly different behavior at low pressures. In contrast, the data by Wang *et al.* show a rapid decrease at much lower pressures.

In order to further support our claim that MIT occurs at $\sim 13\text{-}14 \text{ GPa}$, we measured the infrared (IR) reflectance spectra of $\text{Mn}_3\text{Si}_2\text{Te}_6$ under pressure at room-temperature (Supplementary Fig S12). While a Drude-like mode appears in the low-frequency region of all spectra, recent findings indicate that this rise in the reflectance results from a phononic contribution occurring in the far-IR region below 600 cm^{-1} [arXiv:2311.14673 (2023)]. The reflectivity of $\text{Mn}_3\text{Si}_2\text{Te}_6$ gradually increases with pressure, followed by a sharp rise above $\sim 13 \text{ GPa}$, similar to the transport results. All reflectance spectra were then fitted using a Drude-Lorentz model employed in the ReFIT software [Rev. Sci. Instrum. 76, 083108 (2005)]. The spectra can be well fitted with at least four Lorentz oscillators for $P < 11 \text{ GPa}$, while an additional Drude mode was required to fit the spectra for $P > 11 \text{ GPa}$. The corresponding optical conductivity $\sigma(\omega)$ was calculated via a standard Kramers-Kronig transformation from the fitted reflectance spectra (Fig. S12b). The optical conductivity $\sigma(\omega)$ increases gradually with pressure up to $\sim 13 \text{ GPa}$, followed by a sudden enhancement at low energies indicating the contribution from free charge carriers. Similar behaviour was also observed in VO_2 and attributed to the onset of pressure-induced MIT [Phys. Rev. Lett. 98, 196406 (2007)]. The pressure-dependence of $\sigma(\omega = 600 \text{ cm}^{-1})$, presented in Fig. S12c, exhibits a four-fold enhancement in the optical conductivity above $\sim 13 \text{ GPa}$ (arrow in Fig S12c), consistent with the transport results. This optical conductivity enhancement near P_c provides spectroscopic evidence of the MIT in $\text{Mn}_3\text{Si}_2\text{Te}_6$.

Fig. S12. Infrared spectroscopy studies of $\text{Mn}_3\text{Si}_2\text{Te}_6$ at high pressure. (a) Room- temperature reflectance spectra of $\text{Mn}_3\text{Si}_2\text{Te}_6$ at various pressures. The data between 1600 cm^{-1} and 2700 cm^{-1} are excluded due to strong diamond absorption (indicated by dashed lines). (b) Real part of the complex optical conductivity ($\sigma(\omega)$) at various pressures derived from the fitting to the reflectance spectra. (c) Pressure-dependent optical conductivity at $\omega = 600 \text{ cm}^{-1}$. A four- fold enhancement of the conductivity above $\sim 13 \text{ GPa}$ is indicated by the arrows.

Having established that the transport and IR spectroscopy results on our crystals, along with the recent findings by another independent group [arXiv: 2309.05945 (2023)], consistently reveal the MIT at much higher critical pressure than reported by Wang *et al.* [Phys. Rev. B, 106, 045106 (2022)], we now discuss the possible origins of this discrepancy. Like conventional semiconductors, a small amount of vacancies or impurities in $\text{Mn}_3\text{Si}_2\text{Te}_6$ crystals, incorporated during the synthesis, introduce impurity (acceptor) levels near the valence bands, providing charge carriers, as extensively discussed in Ref. 15 [Nature 599, 576–581 (2021)]. Thus, the temperature-dependent and magnetic-field-dependent resistivity of $\text{Mn}_3\text{Si}_2\text{Te}_6$ single crystals becomes highly sensitive to the impurity concentration within the crystal. The transport properties are mostly dictated by the transport activation gap Δ between the top of the valence bands and the Fermi level pinned by the impurity levels. The higher doping due to a larger amount of impurities indicates the smaller resistivity and magnetoconductivity. Unfortunately, a direct comparison of the resistivity values of our crystals with those of Wang *et al.*'s, is not feasible because they present the resistance of their crystals rather than the resistivity [Phys. Rev. B, 106, 045106 (2022)]. Nevertheless, the magnetoconductivity at low temperatures, which is independent of the geometrical factors of the specimen, can be compared as shown in Fig. R2.

In FIG. R2, we presented the values of magnetoconductivity (MC) ratio at 10 K and 7 T for various $\text{Mn}_3\text{Si}_2\text{Te}_6$ crystals reported in the literature [Phys. Rev. B, 103, L161105 (2021), Phys. Rev. B, 106, 045106 (2022), arXiv: 2309.05945 (2023)]. Typically, the MC ratio at 10 K and 7 T, reflecting the MIT induced by the out-of-plane magnetic field is about 10^6 % or higher. However, the $\text{Mn}_3\text{Si}_2\text{Te}_6$ crystal investigated by Wang *et al.* [Phys. Rev. B, 106, 045106 (2022)] exhibits the smallest MC ratio of $\sim 10^4$ % reported so far, which is at least two orders of magnitude lower than those reported previously, including from this work. The pressure-dependent MC ratio of $\text{Mn}_3\text{Si}_2\text{Te}_6$ crystal grown by Wang *et al.* also differs significantly as compared to the results of our work and the recent report [arXiv: 2309.05945 (2023)], which agree well with each other. Consequently, the low MC ratio and its rapid reduction under pressure suggest that the $\text{Mn}_3\text{Si}_2\text{Te}_6$ crystal grown by Wang *et al.* is highly doped due to a large impurity concentration. These observations raise concerns on the sample quality used in Wang *et al.*'s work, which may mask the intrinsic pressure-dependence of $\text{Mn}_3\text{Si}_2\text{Te}_6$.

FIG. R2. Pressure dependence of magnetoconductivity ratio $\Delta\sigma_{xx}(7\text{T})/\sigma_{xx}(0)$ at 10 K and 7 T of several $\text{Mn}_3\text{Si}_2\text{Te}_6$ crystals reported so far.

Finally, we address a recent theoretical work by Zhang *et al.* [Phys. Rev. B, 107, 054430 (2023)] who identified the critical pressure for the MIT at 2.4 GPa, closely matching the experimental report, by setting the on-site Coulomb interaction of $U = 0.5$ eV. As emphasized in their report, the critical pressure of the MIT highly depends on the value of U , and the choice of $U = 0.5$ eV was specifically made to reproduce the existing experimental findings by Wang *et al.* [Phys. Rev. B, 106, 045106 (2022)]. This means that the calculated critical pressure obtained with $U = 0.5$ eV should not be considered as an independent result supporting the findings of Wang *et al.* Moreover, the value of $U = 0.5$ eV reported in their work appears too small when compared to the U values appropriate for the Mn $3d$ states. In the case of other Mn-based chalcogenides, the typical U values are usually in the range of 3 – 6 eV [Phys. Rev. B., 56, 7222 (1997), Phys. Stat. Sol. (b), 241, 1411 (2004), Phys. Rev. Lett., 123 236401 (2019) and npj Quantum Mater. 5, 56 (2020)]. Indeed, our first-principles calculations demonstrate that the critical pressure of MIT of ~ 13 -14 GPa can be reproduced by considering $U = 5$ -6 eV which falls in the typical range of the typical U values in other Mn chalcogenides.

In conclusion, we have unequivocally demonstrated that the MIT occurs at ~ 13 -14 GPa, supported by our transport measurements on several different crystals as well as IR spectroscopy experiments. This observation is consistent with the recent findings by an independent group [arXiv: 2309.05945 (2023)] confirming that the MIT occurs above 10 GPa, and certainly not between 1.5 - 2.5 GPa as claimed by Wang *et al.* [Phys. Rev. B, 106, 045106 (2022)] based on a single measurement on a single sample.

In the revised Supplementary Information, we have provided additional results of IR measurements under pressure which supports the occurrence of MIT at $P \sim 13$ -14 GPa. We also added discussion on the possible origin for the discrepancy between our work and Wang *et al.*'s work in the revised Supplementary Information.

Q1-2. *The main highlight of this manuscript could be the claim that the pressure-induced transitions occur concomitantly when the nodal-line state crosses the Fermi level in the ferrimagnetic semiconductor Mn₃Si₂Te₆. However, from ref. 22 and the DFT calculations, it is clear that the MIT could be decoupled with both SRT and structural modification. Moreover, the SRT can also be explained by the structural modification.*

A1-2. We thank the reviewer for this comment. In Section **A1-1**, we presented unequivocal evidence establishing the MIT at $P_c \sim 13$ -14 GPa. As previously discussed in the original manuscript, the spin reorientation transition (SRT) and structural modification also occur at the same critical pressure indicating that all three transitions are intimately coupled. To further investigate the possible decoupling of the MIT and SRT, we carried out decompression experiments, during which we measured the transport properties upon releasing pressure from above P_c (Fig. S11). Although a significant hysteresis observed in the pressure dependent resistivity, the MIT behavior is reproduced during decompression. At $P \sim 7.7$ GPa during decompression, we observed the metallic behavior, accompanied by clear hysteresis and a square-shaped field dependence of the anomalous Hall conductivity, indicating the perpendicular magnetic anisotropy. However, after full decompression to $P \sim 0.7$ GPa, the insulating behavior is recovered. Simultaneously, the square-shaped anomalous Hall

Fig. S11. Physical properties of $\text{Mn}_3\text{Si}_2\text{Te}_6$ on decompression from high pressure above P_c . (a) Temperature-dependent ab -plane resistivity $\rho_{ab}(T)$ of $\text{Mn}_3\text{Si}_2\text{Te}_6$ at various pressures during increasing and releasing pressure. Arrows indicate the resistive anomaly at T_c . (b) The magnetoresistance (MR) ratio ($\Delta\rho/\rho(H)$) and (c) Hall conductivity (σ_{xy}) at 15 K measured at various pressures. (d) Pressure-dependent resistivity of $\text{Mn}_3\text{Si}_2\text{Te}_6$ at 300 K showing a significant hysteresis between compression and decompression. (e) Room temperature Raman spectra of $\text{Mn}_3\text{Si}_2\text{Te}_6$ at various pressures. The disappearance of Raman modes on compression above P_c is retained during decompression to $P \sim 1.4$ GPa. In all Figures, label "D" represents data taken on decompression.

conductivity disappears and the magnetoresistivity becomes pronounced. These results suggest that spin direction is no longer along the c -axis and returns to the ab -plane upon full decompression. Notably, the initial structure of $\text{Mn}_3\text{Si}_2\text{Te}_6$ is not entirely recoverable during decompression to ~ 1 GPa as seen by Raman spectroscopy.

These results firmly establish an intimate connection between SRT and MIT. Even with large strain disorder in the decompressed sample leading to significant variation of bond angles and distances between the neighboring Mn atoms significantly from the ideal values, the preferred spin orientation, either along the in-plane or the out-of-plane, is well defined and correlates with the electronic states, either gapped or gapless. Therefore, we can firmly conclude that indeed MIT and SRT are tied together, and the modification of crystal structure could be an additional consequence of the coupled MIT and SRT.

As discussed in the main text, the gap closing between the valence bands with nodal-line degeneracy and the conduction bands is essential for explaining the simultaneous occurrence of the MIT, SRT and structural transition. When the MIT transition is associated with such a gap closing, the charge transfer from the spin-split valence bands with spin-orbit coupling to the conduction bands can occur, which triggers SRT and also induces moderate structural modification. In the case of the high doping as observed by Wang *et al.* [Phys. Rev. B, 106, 045106 (2022)], the MIT transition may occur by increasing the overlap between the impurity bands and the valence bands, without necessarily closing the electronic gap between the valence and conduction bands. Thus, to address the intrinsic pressure-dependent properties of $\text{Mn}_3\text{Si}_2\text{Te}_6$, it is important to use the crystals with lower impurity levels as in our work [Nature 599, 576–581 (2021), this work] and also in recent studies [Phys. Rev. B, 103, L161105 (2021), Phys. Rev. B, 106, 045106 (2022), arXiv: 2309.05945 (2023)].

Q2-1. *In this manuscript, Susilo et. al. studied the transport, magnetic properties, and crystal structure of the ferrimagnetic semiconductor $Mn_3Si_2Te_6$ under high pressures by using the diamond anvil cell. This work reveals that the pressure-induced metal-insulator transition is accompanied with the change of magnetic and crystal structures, undergoing a pressure-induced spin reorientation in the metallic state above the critical pressure P_c . The authors highlight the tuning of pressure on the nodal-line states and the physical properties of $Mn_3Si_2Te_6$. In my opinion, this work is of high quality and deserves publication after addressing the following issues.*

A2-1. We thank the reviewer for his/ her careful reading and for the positive assessment for our manuscript.

Q2-2. *The authors can easily determine the T_c at lower pressures by using the kink feature in resistivity. Can the authors use the same criteria to define T_c above P_c in the metallic state?*

A2-2. We appreciate the insightful comments from the reviewer. As the reviewer pointed out, the resistivity anomaly disappears above P_c in the metallic state, therefore a different criterion was used to determine T_c . As we have shown in the manuscript, once the system enters the metallic state above P_c , the field dependent Hall resistivity $\rho_{xy}(H)$ shows a clear square-shaped behavior indicating the perpendicular magnetic anisotropy above P_c . Since ρ_{xy} is dominated by the anomalous contribution, i.e., $\rho_{xy} \approx \rho_{xy}^A$, the corresponding Hall conductivity, $\sigma_{xy}(H)$, can be scaled nicely with $M(H)$ as shown in Fig. S3. Assuming that the scaling factor $S_H = \frac{\sigma_{xy}}{M}$ is nearly temperature-independent, the net magnetization $M(H)$ can therefore be represented by $\sigma_{xy}(H)$ and thus T_c then can be determined from the temperature dependence of σ_{xy} . This approach enabled us to determine T_c above P_c leading to a complete magnetic phase diagram presented in Fig. 4c of the main text.

To make this part clearer to the reader, we have included relevant sentences in the revised manuscript. We also added the corresponding discussion in Supplementary Note 2.

Fig. S3. Anomalous Hall effect of $Mn_3Si_2Te_6$ above P_c . $M(H)$ curve is nicely reproduced by the field-dependent Hall conductivity $\sigma_{xy}(H)$ with a scaling factor S_H , following a relation $\sigma_{xy}(H) = S_H M(H)$.

Q2-3. The authors stated that the evolution of T_c in this work is consistent with previous reports, e.g. Ref. 26, *Phys. Rev. B* 106, 045106 (2022). However, the behaviors of resistance are very different from that of Ref. 26. What is the cause for such differences?

A2-3. We appreciate the helpful comments from the reviewer. As the reviewer pointed out, there is a clear deviation between our results and those of Wang *et al* [*Phys. Rev. B*, 106, 045106 (2022)]. As described in detail below, we have conducted additional transport experiments on several different crystals, as well as infrared (IR) spectroscopy experiments. All the data consistently show that the metal-insulator transition (MIT) occurs at ~ 13 -14 GPa, which is also in good agreement with the recent work conducted by a different group [arXiv: 2309.05945 (2023)]. They claimed that MIT occurs above 10 GPa and certainly not at $P \sim 1.5$ - 2.5 GPa as claimed by Wang *et al.* based on a single measurement on a single sample [*Phys. Rev. B*, 106, 045106 (2022)].

The critical pressure for the MIT at ~ 13 -14 GPa as determined in our work, relies on measurements for six different $\text{Mn}_3\text{Si}_2\text{Te}_6$ crystals from three different batches. We employed a standard criterion to determine the MIT critical pressure, where the system is inferred to be metallic when the slope of resistivity shows a positive temperature dependence ($d\rho/dT > 0$). The crossover between insulating and metallic resistivity curves is clearly separated by the Mott-Ioffe-Regel (MIR) limit *i.e.* $\rho_{\text{MIR}} = \hbar c/e^2$ (depicted by the dashed line in Fig. R1a) where c is the c -axis lattice constant of $\text{Mn}_3\text{Si}_2\text{Te}_6$. As presented in Fig. R1a, the pressure variation of room-temperature resistivity demonstrates excellent agreement between our different measurements with all data collapsing into a single curve. Additionally, we observed that the pressure-dependence resistivity estimated from the optical conductivity (see below) at low frequency, $\sigma(\omega=600 \text{ cm}^{-1})$, is consistent with the transport results. Furthermore, the recent data by Huang *et al* [arXiv: 2309.05945 (2023)] claimed that MIT occurs above 10 GPa, which agrees well with our work. As shown in Fig. R1b, the pressure-dependence of normalized resistance $\rho_{ab}(P)/\rho_{ab}(\sim 1 \text{ GPa})$ by Huang *et al* [arXiv: 2309.05945 (2023)] follows a similar trend to our results albeit exhibiting a slightly different behavior at low pressures. In contrast, the data by Wang *et al.* only show a rapid decrease at much lower pressures.

FIG. R1. (a) Pressure dependence of room temperature in-plane resistivity of $\text{Mn}_3\text{Si}_2\text{Te}_6$ crystals studied in this work. For comparison, we included the resistivity value estimated from the optical conductivity at low frequency $\sigma(\omega = 600 \text{ cm}^{-1})$ which is consistent with the transport data. Dashed line represents the estimated Mott-Ioffe-Regel limit. (b) Comparison of the pressure-dependent in-plane resistivity, normalized by its value at $P \sim 1$ GPa, for different $\text{Mn}_3\text{Si}_2\text{Te}_6$ crystals.

Fig. S12. Infrared spectroscopy studies of $\text{Mn}_3\text{Si}_2\text{Te}_6$ at high pressure. (a) Room-temperature reflectance spectra of $\text{Mn}_3\text{Si}_2\text{Te}_6$ at various pressures. The data between 1600 cm^{-1} and 2700 cm^{-1} are excluded due to strong diamond absorption (indicated by dashed lines). (b) Real part of the complex optical conductivity ($\sigma(\omega)$) at various pressures derived from the fitting to the reflectance spectra. (c) Pressure-dependent optical conductivity at $\omega = 600\text{ cm}^{-1}$. A four-fold enhancement of the conductivity above $\sim 13\text{ GPa}$ is indicated by the arrows.

In order to further support our claim that MIT occurs at $\sim 13\text{-}14\text{ GPa}$, we measured the infrared (IR) reflectance spectra of $\text{Mn}_3\text{Si}_2\text{Te}_6$ under pressure at room-temperature (Supplementary Fig S12). While a Drude-like mode appears in the low-frequency region of all spectra, recent findings indicate that this rise in the reflectance results from a phononic contribution occurring in the far-IR region below 600 cm^{-1} [arXiv:2311.14673 (2023)]. The reflectivity of $\text{Mn}_3\text{Si}_2\text{Te}_6$ gradually increases with pressure, followed by a sharp rise above $\sim 13\text{ GPa}$, similar to the transport results. All reflectance spectra were then fitted using a Drude-Lorentz model employed in the RefFIT software [Rev. Sci. Instrum. 76, 083108 (2005)]. The spectra can be well fitted with at least four Lorentz oscillators for $P < 11\text{ GPa}$, while an additional Drude mode was required to fit the spectra for $P > 11\text{ GPa}$. The corresponding optical conductivity $\sigma(\omega)$ was calculated via a standard Kramers-Kronig transformation from the fitted reflectance spectra (Fig. S12b). The optical conductivity $\sigma(\omega)$ increases gradually with pressure up to $\sim 13\text{ GPa}$, followed by a sudden enhancement at low energies indicating the contribution from free charge carriers. Similar behaviour was also observed in VO_2 and attributed to the onset of pressure-induced MIT [Phys. Rev. Lett. 98, 196406 (2007)]. The pressure-dependence of $\sigma(\omega = 600\text{ cm}^{-1})$, presented in Fig. S12c, exhibits a four-fold enhancement in the optical conductivity above $\sim 13\text{ GPa}$ (arrow in Fig S12c), consistent with the transport results. This optical conductivity enhancement near P_c provides spectroscopic evidence of the MIT in $\text{Mn}_3\text{Si}_2\text{Te}_6$.

Having established that the transport and IR spectroscopy results on our crystals, along with the recent findings by another independent group [arXiv: 2309.05945 (2023)], consistently reveal the MIT at much higher critical pressure than reported by Wang *et al.* [Phys. Rev. B, 106, 045106 (2022)], we now discuss the possible origins of this discrepancy. Like conventional semiconductors, a small amount of vacancies or impurities in $\text{Mn}_3\text{Si}_2\text{Te}_6$ crystals, incorporated during the synthesis, introduce impurity (acceptor) levels near the valence bands, providing charge carriers, as extensively discussed in Ref. 15 [Nature 599, 576–581 (2021)].

FIG. R2. Pressure dependence of magnetoconductivity ratio $\Delta\sigma(7T)/\sigma(0)$ at 10 K and 7 T of several $\text{Mn}_3\text{Si}_2\text{Te}_6$ crystals reported so far.

Thus, the temperature-dependent and magnetic-field-dependent resistivity of $\text{Mn}_3\text{Si}_2\text{Te}_6$ single crystals becomes highly sensitive to the impurity concentration within the crystal. The transport properties are mostly dictated by the transport activation gap Δ between the top of the valence bands and the Fermi level pinned by the impurity levels. The higher doping due to a larger amount of impurities indicates the smaller resistivity and magnetoconductivity. Unfortunately, a direct comparison of the resistivity values of our crystals with those of Wang *et al.*'s, is not feasible because they present the resistance of their crystals rather than the resistivity [Phys. Rev. B, 106, 045106 (2022)]. Nevertheless, the magnetoconductivity at low temperatures, which is independent of the geometrical factors of the specimen, can be compared as shown in Fig. R2.

In FIG. R2, we presented the values of magnetoconductivity (MC) ratio at 10 K and 7 T for various $\text{Mn}_3\text{Si}_2\text{Te}_6$ crystals reported in the literature [Phys. Rev. B, 103, L161105 (2021), Phys. Rev. B, 106, 045106 (2022), arXiv: 2309.05945 (2023)]. Typically, the MC ratio at 10 K and 7 T, reflecting the MIT induced by the out-of-plane magnetic field is about 10^6 % or higher. However, the $\text{Mn}_3\text{Si}_2\text{Te}_6$ crystal investigated by Wang *et al.* [Phys. Rev. B, 106, 045106 (2022)] exhibits the smallest MC ratio of $\sim 10^4$ % reported so far, which is at least two orders of magnitude lower than those reported previously, including our work. The pressure-dependent MC ratio of $\text{Mn}_3\text{Si}_2\text{Te}_6$ crystal grown by Wang *et al.* also differs significantly as compared to the results of our work and the recent report [arXiv: 2309.05945 (2023)], which agree well with each other. Consequently, the low MC ratio and its rapid reduction under pressure suggest that the $\text{Mn}_3\text{Si}_2\text{Te}_6$ crystal grown by Wang *et al.* is highly doped due to a large impurity concentration. These observations raise concerns on the sample quality used in Wang *et al.*'s work, which may mask the intrinsic pressure dependence of $\text{Mn}_3\text{Si}_2\text{Te}_6$.

Q2-4. As described in the methods, the authors have used different pressure transmitting media for different high-pressure measurements. However, the results from different measurements seem to be quite consistent. Are there any influence of different medium on the physical properties? If so the authors may discuss briefly.

A2-4. We thank the reviewer for this comment. As the reviewer mentioned, the use of different

pressure media introduces varied and potentially severe stresses on the sample, particularly under non-hydrostatic conditions, where the sample experiences distinct stresses along different crystal directions. In some cases, this might affect the properties highly sensitive to stress gradient such as superconductivity. In general, the non-hydrostatic conditions are more likely to promote earlier or lower pressure-induced transitions. We agree with the reviewer that our results on $\text{Mn}_3\text{Si}_2\text{Te}_6$ are consistent although different pressure media were used. This indicates that $\text{Mn}_3\text{Si}_2\text{Te}_6$ is not highly sensitive to the non-hydrostatic condition. We note that such behavior is not unusual even in the case of layered materials where the interlayer spacing is important, as observed in SnS_2 [RSC Adv., 12, 2454 (2022)] and CrCl_3 [Inorg. Chem., 61, 4852 (2022)].

Following the reviewer's suggestion, we included related sentences in the revised manuscript.

Q2-5. The authors stated that a pressure-driven spin reorientation appears across the P_c , which can be extracted from the $M(H)$ curves with obvious hysteresis loop for $H//c$ at higher pressures above 12GPa. While for $H//ab$, the $M(H)$ is only measured to 7.9GPa, as is shown in Fig.2. Why the authors only increase the pressure to 7.9GPa? They should provide the $M(H)$ curves at higher pressures to further confirm the pressure-induced spin reorientation with obvious reduction of magnetization for $H//ab$.

A2-5. We appreciate this comment from the reviewer. The majority of the magnetization data for $\text{Mn}_3\text{Si}_2\text{Te}_6$ were obtained using a modified turnbuckle diamond anvil cell (DAC) based on the design by Giriat et al. [Rev. Sci. Instrum., 81, 073905 (2010)]. The dimensions of the sample space in the commercial Quantum Design SQUID MPMS magnetometer is approximately 10 mm long and 7 mm in diameter, which is compatible with the DAC for $H//c$ configuration, where the magnetic field parallel to the direction of the applied load. To measure $M(H)$ for $H//ab$, however, we had to use a much smaller DAC in length to rotate it by 90 degrees inside the sample space, which is more fragile at high pressures. This is the main reason why we only measured $M(H)$ for $H//ab$ up to ~ 8 GPa.

We made a new attempt to increase pressure close to $P_c \sim 13$ -14 GPa, however the body of the pressure cell was not able to withstand the high load and broke instead. As shown in Fig R3, our new experiments only reached a maximum pressure of ~ 8.5 GPa, consistent with our previous results presented in Figs. 2(h) and (i) of the original manuscript, Supplementary Note 3 and Supplementary Fig S4. We note that although many types of DAC have been designed

FIG. R3. (a-c) Additional field-dependent magnetization ($M(H)$) of $\text{Mn}_3\text{Si}_2\text{Te}_6$ measured up to 8.5 GPa. Arrows indicate the saturation magnetic field. (d) Pressure-dependent of magnetocrystalline anisotropy energy K . (e) Pressure-dependent saturation magnetization M_{sat} and the low-field c -axis magnetization M_c for $H//c$ at 5 K.

and employed for magnetic measurements at high pressures and low temperatures [Rev. Sci. Instrum., 82, 053906 (2011), Rev. Sci. Instrum., 86, 093901 (2015), High Press. Res., 37, 465 (2017)], all of them are typically capable of measuring magnetization with the magnetic field parallel to the direction of applied load. To the best of our knowledge, magnetic measurements with applied magnetic field perpendicular to the direction of load, as reported in this work, are quite rare.

Due to the technical limitation in measuring field-dependent magnetization for $H//ab$ above $P \sim 9$ GPa, we turned to the anomalous Hall conductivity data ($\sigma_{xy}(H)$) which shows a clear square-shaped field-dependence. This allowed us to show that the easy-axis of magnetization is no longer in the ab -plane, but instead along the c -axis above P_c . The square-shaped behavior in $\sigma_{xy}(H)$ along the magnetic easy-axis is indeed common and have been reported in several topological magnets [Nature, 527, 212 (2015), Phys. Rev. Applied, 5, 064009 (2016), Nature Phys 14, 1125–1131 (2018), Phys. Rev. Mater., 4, 044203 (2020), Nature 583, 533–536 (2020)]. Therefore, we believe that a clear square-shaped hysteresis loop of $\sigma_{xy}(H)$ with $H//c$, together with a clear jump in the field-dependent magnetization for $H//c$, provide sufficient evidence to justify that the Mn^{2+} spin orientation is along the c -axis above P_c .

Q2-6. *As indicated by the high-pressure XRD patterns, a structural transformation appears at the P_c from trigonal structure to the monoclinic structure. However, the authors discuss the changes of electronic structures with the same crystal structure below and above the P_c .*

A2-6. We thank the reviewer for pointing out this issue, which was probably not clear in the original manuscript. As the reviewer mentioned the electronic structures of the low-pressure semiconducting trigonal phase should indeed differ from the high-pressure semimetal monoclinic phases. However, near the critical pressure P_c approaching from the lower pressure in the triclinic semiconducting phase, the nodal-line states in the valence bands at the Γ point becomes closer to the Fermi level. This proximity triggers the MIT, accompanied by the SRT and the structural modification as discussed in the main text. Thus, our discussion regarding the possible origin for the concomitant SRT and structural modification at P_c is based on the pressure-driven changes of the electronic structure in the low-pressure trigonal phase.

Nevertheless, we can find the signature of the nodal-line states even in the electronic structure of the semimetal monoclinic phase, just above P_c . As the MIT occurs at $P_c \sim 13$ -14 GPa, we

FIG. R4. Electronic band structures of $Mn_3Si_2Te_6$ at 14 GPa for the high pressure crystal structure ($C2/c$ space group) calculated with $U = 5$ eV.

conducted geometry optimization on the crystal structures at $P = 14$ GPa and calculated band structures at various values of the on-site Coulomb energy (U) without spin-orbit coupling. Selecting $U = 5$ eV allowed us to reproduce the band gap closure around $P_c \sim 12$ GPa on both the trigonal and monoclinic structures. Figure R4 depicts the electronic structures just above P_c obtained using DFT calculations with $U = 5$ eV in the high-pressure monoclinic phase. Notably the nodal-line states persist at the Γ point and crosses the Fermi level (E_F). Upon further increasing pressure far above P_c , we expect that other trivial bands will be introduced at the E_F , weakening the effect of nodal-line states on the physical properties. While the possible role of the nodal-line states well above P_c is an interesting question, we think it is beyond the scope of this work.

Q2-7. *As reported in the previous report Phys. Rev. B 106, 045106 (2022), the high-pressure phase can be quenched to lower pressures. Have the authors measured the transport properties, magnetization and high-pressure XRD in the decompression process? If the transition is irreversible, the authors can measure the magnetic properties at even ambient pressure and can further confirm the observed results.*

A2-7. We really appreciate this valuable comment from the reviewer. To address this concern, we have performed an additional run with a focus of investigating the transport and vibrational properties on decompression from above P_c as shown in Supplementary Fig. S11. The transport measurements reveal that the metal-insulator transition (MIT) is reversible. In particular, the in-plane resistivity increases sharply by more than two orders of magnitude, accompanied by strong insulating behavior as the pressure is reduced to below ~ 2 GPa. This observation indicates that the band gap re-opens and the semiconducting state of $\text{Mn}_3\text{Si}_2\text{Te}_6$ is recovered upon releasing pressure.

In addition, the magnetoresistivity (MR) ratio, which is negligible in the metallic state above P_c , gradually increases with decreasing pressure and becomes pronounced in the insulating state at ~ 0.7 GPa. Simultaneously, the large Hall conductivity $\sigma_{xy}(H)$ in the metallic state is weakened and becomes negligible below ~ 1 GPa. A negligible $\sigma_{xy}(H)$ in the insulating state suggests that the spin direction is no longer along the c -axis and returns to the ab -plane on decompression. These observations are entirely opposite to the behavior during compression, confirming the reversibility of $\text{Mn}_3\text{Si}_2\text{Te}_6$, albeit with a large hysteresis. This large hysteresis is evident in the room temperature Raman spectra, where the initial Raman modes observed below P_c are not fully recoverable during decompression to ~ 1 GPa.

These results firmly establish that the SRT is intimately connected to MIT. Even with large strain disorder in the decompressed sample, leading to variation of bond angles and distances between the neighboring Mn atoms that deviate significantly from the ideal values, the preferred spin orientation, either along the in-plane or the out-of-plane directions, is well defined and correlated to the electronic states, either gapped or gapless. Therefore, it can be concluded that indeed MIT and SRT are tied together, and crystal structure modification at P_c could be an additional consequence of the coupled MIT and SRT.

We have included relevant discussions about decompression behavior of $\text{Mn}_3\text{Si}_2\text{Te}_6$ in Supplementary Note 9.

Fig. S11. Physical properties of $\text{Mn}_3\text{Si}_2\text{Te}_6$ on decompression from high pressure above P_c . (a) Temperature-dependent ab -plane resistivity $\rho_{ab}(T)$ of $\text{Mn}_3\text{Si}_2\text{Te}_6$ at various pressures during increasing and releasing pressure. Arrows indicate the resistive anomaly at T_c . (b) The magnetoresistance (MR) ratio ($\Delta\rho/\rho(H)$) and (c) Hall conductivity (σ_{xy}) at 15 K measured at various pressures. (d) Pressure-dependent resistivity of $\text{Mn}_3\text{Si}_2\text{Te}_6$ at 300 K showing a significant hysteresis between compression and decompression. (e) Room temperature Raman spectra of $\text{Mn}_3\text{Si}_2\text{Te}_6$ at various pressures. The disappearance of Raman modes on compression above P_c is retained during decompression to $P \sim 1.4$ GPa. In all Figures, label "D" represents data taken on decompression.

List of the changes made

1. We included two additional authors, Philip Dalladay-Simpson and Zifan Wang, who performed high pressure infrared (IR) reflectance spectroscopy measurements and additional band structure calculations.

2. Considering the comments from Reviewers #1 and #2 about the deviations from the recent report by Wang *et al.* [Phys. Rev. B, 106, 045106 (2022)] in **Q1-1** and **Q2-3**, we have provided additional results of IR reflectance measurements under pressure in the revised Supplementary Information, which supports the occurrence of MIT above ~ 13 -14 GPa. We also added a short discussion about the possible origin for the discrepancy between our work and Wang *et al.*'s work in the revised Supplementary Information.

In the main text:

[page 4-5] We added the following sentences: “The similar behavior were observed for six different crystals from ... , where c is the c -axis lattice constant of $\text{Mn}_3\text{Si}_2\text{Te}_6$.” and “The MIT at $P_c \sim 14$ GPa is further confirmed by infrared (IR) reflectance spectroscopy, ... , most likely due to different doping levels in $\text{Mn}_3\text{Si}_2\text{Te}_6$ crystals (Supplementary Note 11).”

In Supplementary Information:

[Supplementary Note 10] New supplementary note about the additional results of IR reflectance measurements under pressure.

[Supplementary Figure S12] New results of IR reflectance measurements under pressure.

[Supplementary Note 11] New supplementary note discussing about the possible origin for the discrepancy between our work and Wang *et al.*'s work.

[Supplementary Figure S13] Comparison of transport data for several $\text{Mn}_3\text{Si}_2\text{Te}_6$ crystals reported so far is presented.

3. Considering the comments from Reviewer #2 about the decompression behavior and from Reviewer #1 about the possibility of decoupling between MIT and SRT in **Q1-2** and **Q2-7**, we have added additional transport and Raman spectroscopy measurements on releasing pressure from above P_c in the revised Supplementary Information.

In the main text:

[page 10] We added the following sentences: “Furthermore, transport and Raman spectroscopy measurements taken on decompression... concluded that indeed the MIT and SRT are tied together due to the SOC effect in the nodal-line states.”

In Supplementary Information:

[Supplementary Note 9] New supplementary note about the additional transport and Raman spectroscopy measurements measured on releasing pressure from above P_c .

[Supplementary Figure S11] New results of decompression behavior of $\text{Mn}_3\text{Si}_2\text{Te}_6$.

4. Considering the comment from Reviewer #2 about the criterion used to determine T_c of $\text{Mn}_3\text{Si}_2\text{Te}_6$ above P_c in Q2-2, we have provided additional discussion in the revised main text and Supplementary Note.

In the main text:

[pages 6 - 7] We added the following sentences: “The $M(H)$ data can be nicely reproduced by $\sigma_{xy}(H)$ following a relation $\sigma_{xy}(H) = S_H M(H)$ (Supplementary Note 2).”

In Supplementary Information:

[Supplementary Note 2] New discussion about the anomalous Hall response above P_c is added.

[Supplementary Figure S3] A new figure which describes the relationship between $\sigma_{xy}(H)$ and $M(H)$ is added.

5. Considering the comment from Reviewer #2 about the influence of different pressure medium used in our work in Q2-4, we have included a brief discussion in the revised manuscript.

In the main text:

[page 12] We added the following sentences: Despite the use of different pressure medium between different set of experiments, ... $\text{Mn}_3\text{Si}_2\text{Te}_6$ is not sensitive to the different hydrostatic conditions.”

6. Minor changes are listed below.

In Supplementary Information:

[References] A new reference added. “Q. Wu et al., Pump-induced terahertz conductivity response and peculiar bound state in $\text{Mn}_3\text{Si}_2\text{Te}_6$. *arXiv:2311.14673* (2023).”

[References] A new reference added. “A. B. Kuzmenko, Kramers–Kronig constrained variational analysis of optical spectra. *Rev. Sci. Instrum.* 76, 083108 (2005).”

[References] A new reference added. “E. Arcangeletti et al., Evidence of a Pressure-Induced Metallization Process in Monoclinic VO_2 , *Phys. Rev. Lett.* 98, 196406 (2007).”

[References] A new reference added. “C. Huang et al., Gap and magnetic engineering via doping and pressure in tuning the colossal magnetoresistance in $(\text{Mn}_{1-x}\text{Mg}_x)_3\text{Si}_2\text{Te}_6$. *arXiv:2309.05945* (2023).”

Reviewers' Comments:

Reviewer #1:

Remarks to the Author:

The authors have improved their manuscript in the revised version, but have not addressed my concerns.

In my previous report, one of my main concerns is the critical pressure of the metal-insulator transition (MIT). Based on their data, the authors claimed that the MIT occurs at ~ 13 - 14 GPa, which is much higher than 1.5 - 2.5 GPa obtained from both a previous experimental measurement (ref. 22) and a theoretical work (PHYSICAL REVIEW B 106, 045106 (2022)).

In the response to reviewers' comments, they further cited a recent work (arXiv: 2309.05945 (2023)) to support their arguments. In the first paragraph of A1-1 in Response to Reviewer #1's comments, one can read: "They claimed that MIT occurs above 10 GPa and certainly not at $P \sim 1.5 - 2.5$ GPa as claimed by Wang et al. based on a single measurement on a single sample [Phys. Rev. B, 106, 045106 (2022)]." However, this is not the truth. The exact words in arXiv: 2309.05945 (2023) are: "The gap closes and a MIT occurs at ~ 6 GPa, where the CMR is abruptly reduced."

Although the critical pressure of the MIT might rely on the sample quality, I still believe that the MIT is decoupled with both the spin-reorientation transition (SRT) and structural modification. The decoupled transitions not only can be deduced from both ref. 22 and the DFT calculations, but also are presented in the phase diagram in the recent work performed by Huang et al. (arXiv: 2309.05945 (2023)).

Reviewer #2:

Remarks to the Author:

The authors have addressed properly my concerns and I recommend accepting the revision for publication.

Q1-1. *The authors have improved their manuscript in the revised version, but have not addressed my concerns. In my previous report, one of my main concerns is the critical pressure of the metal-insulator transition (MIT). Based on their data, the authors claimed that the MIT occurs at ~ 13-14 GPa, which is much higher than 1.5–2.5 GPa obtained from both a previous experimental measurement (ref. 22) and a theoretical work (PHYSICAL REVIEW B 106, 045106 (2022)).*

In the response to reviewers' comments, they further cited a recent work (arXiv: 2309.05945 (2023)) to support their arguments. In the first paragraph of A1-1 in Response to Reviewer #1's comments, one can read: "They claimed that MIT occurs above 10 GPa and certainly not at $P \sim 1.5 - 2.5$ GPa as claimed by Wang et al. based on a single measurement on a single sample [Phys. Rev. B, 106, 045106 (2022)]." However, this is not the truth. The exact words in arXiv: 2309.05945 (2023) are: "The gap closes and a MIT occurs at ~6 GPa, where the CMR is abruptly reduced." Although the critical pressure of the MIT might rely on the sample quality, I still believe that the MIT is decoupled with both the spin-reorientation transition (SRT) and structural modification. The decoupled transitions not only can be deduced from both ref. 22 and the DFT calculations, but also are presented in the phase diagram in the recent work performed by Huang et al. (arXiv: 2309.05945 (2023)).

A1-1. We appreciate the helpful comments from the reviewer. In response to the reviewer's concerns, we offer a comprehensive discussion on several key issues below, including (1) ensuring consistency between the results presented in Huang *et al.* [arXiv: 2309.05945 (2023)] and our work, (2) establishing criteria for determining the critical pressure (P_c) based on the magnetotransport properties, and (3) validating the calculated P_c from DFT calculations. Let us address each of these issues one by one.

(1) Consistency between the results of Huang *et al.*'s and our works.

As the reviewer correctly pointed out, Huang *et al.* mentioned in their manuscript [arXiv: 2309.05945 (2023)] that the MIT occurs at ~6 GPa. However, upon careful examination, we have found that their experimental results are in excellent agreement with ours, strongly suggesting that MIT takes place at pressures higher than 10 GPa. On closer examination of their resistance data, it can be seen that the normalized resistance curves of their $\text{Mn}_3\text{Si}_2\text{Te}_6$ crystal at $P = 6.5, 8.0$ and 10.0 GPa consistently exhibit a negative temperature dependence ($d\rho/dT < 0$) across the whole temperature range (FIGS. R1a and j below), indicating its semiconducting state even at $P = 10$ GPa. These behaviors are almost identical to the normalized resistance data in our work for two different crystals at pressures ranging $P = 6 - 12$ GPa (FIGS. R1 k-l).

Using the resistance data extracted from Fig. 4b in Huang *et al.*'s work, we found that their resistivity curves follow the activation behavior even at $P = 10$ GPa, as clearly shown in the Arrhenius plot (FIGS. R1b-h). The activation gap, estimated from the Arrhenius plot, is systematically reduced with pressure but remains finite at high pressures, for example, ~5 meV at 10 GPa (green solid triangles in FIG. R1i). The pressure-dependent gap data, obtained from Huang *et al.*'s results, matches well with those from our three crystals (open symbols in FIG. R1i). This detailed comparison reveals the consistency between the results in Huang *et al.* and

FIG. R1. (a) In-plane resistivity of $\text{Mn}_3\text{Si}_2\text{Te}_6$ between 6 – 10 GPa adapted from Huang *et al.*'s work [arXiv: 2309.05945 (2023)]. (b - h) Arrhenius plot of the resistance data. Solid lines are fits to the data in the temperature range between 250 K – 300 K. (i) Pressure-dependent activation gap data taken from the resistivity data (a) from Huang *et al.*'s. The gap data correspond to those presented in the manuscript (solid purple circle) and obtained by the analysis shown in (b-h) (solid green triangles). The activation gap data estimated for three crystals in our work are plotted together (open symbols) for comparison. (j - l) Comparison of normalized resistivity of $\text{Mn}_3\text{Si}_2\text{Te}_6$ crystals from Huang *et al.*'s work and our works for the pressure range 6 – 12 GPa.

our work, unequivocally indicating that the metal-insulator transition in $\text{Mn}_3\text{Si}_2\text{Te}_6$ occurs above 10 GPa, rather than at $P \sim 6$ GPa.

(2) Criterion for the critical pressure (P_c) based on the magnetotransport properties

In FIGS. R2a, b, we show the comparison of pressure dependences of magnetoresistance ($\text{MR} = \frac{(\rho_{xx}(H) - \rho_{xx}(0))}{\rho_{xx}(0)}$) and magnetoconductivity ($\text{MC} = \frac{(\sigma_{xx}(H) - \sigma_{xx}(0))}{\sigma_{xx}(0)}$) values of $\text{Mn}_3\text{Si}_2\text{Te}_6$ crystals reported by Huang *et al.* and in our work. Again, both MR and MC results in Huang *et al.* and our work exhibit good agreement with each other.

As the referee quoted, Huang *et al.* consider the abrupt change in MR at ~ 6 GPa as a signature of the metal-insulator transition. However, it has been well known that the MR data should be interpreted carefully when determining the phase transition, particularly for colossal magnetoresistive materials, like $\text{Mn}_3\text{Si}_2\text{Te}_6$. In $\text{Mn}_3\text{Si}_2\text{Te}_6$, the zero-field resistivity $\rho_{xx}(0)$ at

FIG. R2. Comparison of pressure dependence of (a) magnetoresistance $(\rho_{xx}(H) - \rho_{xx}(0))/\rho_{xx}(0)$ and (b) magnetoconductivity $(\sigma_{xx}(H) - \sigma_{xx}(0))/\sigma_{xx}(0)$ at 10 K for $\text{Mn}_3\text{Si}_2\text{Te}_6$ crystals reported in this work and by Huang *et al* [arXiv: 2309.05945 (2023)].

low temperature ($T = 2$ K) is in the order of $10^6 - 10^7 \Omega \text{ cm}$, which drops 7 – 8 orders of magnitude, reaching $\rho_{xx}(H) \sim 10^{-1} \Omega \text{ cm}$ under magnetic fields larger than 4 T [Phys. Rev. B, 103, L161105 (2021), Nature 599, 576–581 (2021), arXiv: 2309.05945 (2023)]. In such a case, the *MR* ratio, as defined above, is nearly equal to -100% in the *linear* scale. While the $\rho_{xx}(H)$ monotonically and exponentially decreases with increasing pressure, the change in *MR* ratio in the *linear* scale becomes apparent only when $\rho_{xx}(H)$ is comparable to $\rho_{xx}(0)$. The corresponding magnetoconductivity in the *log* scale, which can capture such a substantial resistivity change with magnetic fields, exhibits no distinctive anomaly in both our and Huang *et al*'s results (FIG. R2b). Therefore, the seemingly sharp increase of the *MR* at ~ 6 GPa in the *linear* scale is a consequence of the exponential and monotonic pressure dependence of magnetoconductivity, which should not be interpreted as a signature of metal-insulator transition.

(3) Estimation of P_c using DFT calculations

As we already emphasized our previous reply, the critical pressure for the metal-insulator transition in $\text{Mn}_3\text{Si}_2\text{Te}_6$ is sensitive to the choice of the on-site Coulomb interaction strength (U). In a recent theoretical study by Zhang *et al.* [Phys. Rev. B, 107, 054430 (2023)], the on-site Coulomb interaction of $U = 0.5$ eV was assumed to reproduce the experimental $P_c \sim 2.5$ GPa, reported by Wang *et al.* [Phys. Rev. B, 106, 045106 (2022)], which was the only available data at that time. We note that the assumed $U = 0.5$ eV appears relatively small compared to the typical U values of $\sim 3 - 6$ eV for other Mn-based chalcogenides [Phys. Rev. B., 56, 7222 (1997), Phys. Stat. Sol. (b), 241, 1411 (2004), Phys. Rev. Lett., 123 236401 (2019) and npj Quantum Mater. 5, 56 (2020)].

In our previous reply, we indeed showed that by choosing $U = 5$ eV in our DFT calculations yield the critical pressure for MIT to be $P_c \sim 12$ GPa, consistent with our experimental findings. In order to further support our claim on spin reorientation at P_c , we have calculated the total energy difference between spin orientations along [110] and [001] directions ($\Delta E = E[110] - E[001]$) using the structural information determined by X-ray diffraction experiments at different pressures. At ambient pressure, the negative ΔE corresponds to the preferred spin

FIG. R3. Calculated pressure dependences of total energy difference between $M//ab$ ($E[110]$) and $M//c$ ($E[001]$) i.e. $\Delta E = E[110] - E[001]$ based on LDA+U+SOC ($U = 5$ eV). The lattice parameters are fixed to the experimental values at each pressure.

direction in the ab -plane. With increasing pressure, ΔE approaches to zero and eventually becomes positive above ~ 14 GPa. These results indicate the change in the preferred spin orientation to the $[001]$ direction, once the gap closes at MIT, consistent with our experimental results.

While additional theoretical investigations are required for a comprehensive understanding of magnetic anisotropy and its pressure dependence, our calculations clearly show that a more appropriate choice of the on-site Coulomb energy U allows us to reproduce both experimentally-identified metal-insulator and spin orientation transitions at high pressures.

As discussed above, the reviewer's concerns regarding a few issues in the previous studies can be resolved by careful examination on the experimental data and additional calculations.

Lastly, we would like to emphasize again that, in addition to the aforementioned results, we have provided several experimental findings supporting our claims. In the previous response and revised manuscript, we have shown that the MIT occurs at $P_c \sim 14$ GPa, based on our six measurements on three different crystals as well as IR spectroscopy experiments (Supplementary Notes 10 and 11), which is also further supported by the recent work by Huang *et al.* [arXiv: 2309.05945 (2023)] discussed above. The critical pressure at which the MIT occurs matches well with the SRT, observed from magnetization experiments. Moreover, in the previous revised manuscript, we have provided a critical finding from decompression experiments. Specifically, we observed that the square-shaped anomalous Hall conductivity, a signature of the perpendicular magnetic anisotropy, disappears as the insulating state of $\text{Mn}_3\text{Si}_2\text{Te}_6$ is recovered on releasing pressure below 1 GPa, establishing an intimate connection between SRT and MIT. This observation implies that the preferred spin orientation, either along the in-plane or the out-of-plane, is well defined and correlates with the electronic states, either gapped or gapless (Supplementary Note 9). All these experimental evidences unequivocally support our claim and firmly establishes that MIT and SRT are coupled together and occur simultaneously at high pressures.